# Road Impact on Plant Colonization in the Arid Timanfaya National Park

**DOI:** 10.3390/plants12203568

**Published:** 2023-10-13

**Authors:** María Bernardos, Natalia Sierra Cornejo, Antonio Daniel Torres Hassan, Raimundo Cabrera, José Ramón Arévalo

**Affiliations:** 1Departamento de Botánica, Ecología y Fisiología Vegetal, Facultad de Biología, Universidad de La Laguna, 38206 La Laguna, Spain; mariabernardoshernandez@gmail.com (M.B.); nsierrac@ull.edu.es (N.S.C.); rcabrera@ull.edu.es (R.C.); 2Gabinete de Estudios Ambientales (GEA), Single-Member Limited Company, 38591 Santa Cruz de Tenerife, Spain; tonith93@gmail.com

**Keywords:** arid protected areas, colonization, primary succession, richness, road-edge effect, species composition

## Abstract

Roads have the potential to alter local environmental conditions, such as the availability of water and nutrients, and rapidly create suitable habitats for the establishment of both native and non-native plant species, transforming the ecosystems. This is a challenge in Timanfaya National Park and Los Volcanes Natural Park on Lanzarote Island, protected areas that have experienced primary succession after recent volcanic eruptions. In arid ecosystems, changes in abiotic conditions along roadsides might facilitate colonization and plant growth. We analyzed the effect of roads and road type on plant species composition and richness at a spatiotemporal scale. Vascular plant species were systematically recorded at three distances from the road edge on both sides, across fourteen zones in the wet and dry seasons, for three years. Results showed that there were slight differences on species composition depending on the distance to the road edge, as well as on the zones. Species richness was also determined by the interaction of the position, zones, and season, being higher at the road edge. Furthermore, zones with higher traffic intensity showed a higher presence of both native and non-native species. This study highlights the importance of the awareness about the road impacts on species composition by enhancing the colonization capacity of species while facilitating the entry of invasive ones. Good management practices regarding infrastructures in natural protected areas are crucial for the conservation of their unique flora, landscapes, and natural succession processes.

## 1. Introduction

The presence of touristic destinations and the associated road network has several effects on the surrounding landscape [1,2]. Roads generate collateral problems for ecosystem conservation and landscape integrity [3,4,5,6,7], such as habitat fragmentation [8,9] and rapid native and non-native species colonization. Roads and trails act as dispersal corridors for both native and alien species [10,11,12,13], especially those with short life cycles and high reproductive rates [14]. As a result, seeds are easily dispersed by vehicles, hikers, and domestic animals and deposited due to vehicle-induced air turbulence or favorable roadside conditions [15]. Both roads and trails can alter local environmental conditions, promoting the establishment of native and non-native plants adapted to disturbance [16]. Compared to roads, trails are less susceptible to invasions due to lower propagule pressure and impact severity [17,18]. Additionally, roads create unique environmental conditions, including microclimates and soil characteristics, which make them attractive to native and non-native species [19,20]. They can cause changes through variations in solar radiation, wind regimes, moisture, and temperature [21,22]. Road verges offer specific environmental conditions, such as varied sun and shade combinations, different slopes, and different soil conditions, which contribute to roadside species richness [15]. Microclimate gradients across road edges can transform ecosystems into suitable habitats for plants and animals, thereby affecting ecological processes and patterns [21,22]. The road network can promote the colonization and persistence of alien species with broad environmental ranges [2,23,24], contributing to their spread [2,25] and causing local extinctions of native species and the homogenization of the resident communities. When roads are built, the soil undergoes various alterations that can further impact plant colonization. Additionally, the construction of roads often leads to the generation of embankments and clearings that create new opportunities for plant colonization. The exposed slopes and leveled areas provide suitable habitats for plant growth, allowing species to take root and establish themselves more easily. However, during this colonization process, exotic species tend to exhibit greater aggressiveness compared to native species. They can outcompete native species for resources and quickly colonize these disturbed areas, potentially leading to the displacement of indigenous vegetation. The road-edge effect extends beyond just a few meters but can influence the surrounding habitats for hundreds of meters; for instance, in terms of the plant species composition and vegetation structure [26].

Oceanic islands worldwide are being heavily changed and constrained by urbanization and transportation pressures [27,28]. Over the past four decades, the road network in the Canary Islands, including the touristic Lanzarote Island, has experienced rapid growth, making it one of the most heavily roaded regions in Europe [29,30]. This road network’s expansion and the resulting increase in road edges further exacerbate the proliferation of native and non-native species, including invasive ones [28,31]. Dispersal of plants along roads may accelerate plant invasions, inducing rapid changes in biodiversity patterns, especially on road verges, where the disturbance effect of the anthropogenic corridor is more evident [30,32,33]. In dryland ecosystems such as Lanzarote, where water availability is the main limiting factor, roads can have significant effects on water redistribution [34]. Another limiting factor which can pose challenges for plant colonization is the composition of the substrate, particularly in the case of volcanic deposits or lava flows. These types of substrates often exhibit specific physical and chemical characteristics, such as high acidity, nutrient deficiency, or the presence of toxic substances, that can impede plant growth and restrict the diversity and abundance of plant species [4].

In the case of Timanfaya National Park and Los Volcanes Natural Park, the ecological succession process is still ongoing, since the eruptions occurred relatively recently in geological terms (between 1730 and 1736 and 1824). These eruptions, that took place during the mentioned time periods, resulted in the formation of extensive lava fields, volcanic cones, craters, and other volcanic landforms that characterize the landscape of Lanzarote today. These parks hold significant geological value and exhibit a considerable variability of environments. Within these areas, there are regions characterized by different climatic conditions, such as windier and colder areas, as well as areas with higher humidity and warmer temperatures. This environmental variability can have diverse effects on the establishment of species, and consequently, on species composition in different zones; if roads are also present, the environments can be altered in different ways. Moreover, maybe because of the aggressive use of these protected areas for tourism (1,629,255 visitors in 2019 [35]), plant invasions are likely to increase, destroying native habitats and creating new habitats for species adapted to human-disturbed environments [36].

To understand the influence of zones on particular environmental conditions, we studied vascular plant species at various distances from the road edge, encompassing different zones within the protected areas. Additionally, data collection occurred during both the wet and dry seasons over a three-year period. By considering the specific characteristics of the study area and the seasonal dynamics, this research aimed to provide a comprehensive understanding of the complex relationship between the effect of roads and zones with different environmental conditions on species composition and richness. Such knowledge is essential for informed decision-making, effective management practices, and the preservation of the unique ecosystems.

We expect a clear road-edge effect that leads to greater species richness and compositional differences between the border and interior of these protected areas. This effect is estimated to be more pronounced in zones with higher anthropogenic pressure compared to more inaccessible ones, as well as in zones located further to the north (with greater moisture input) compared to those situated further south. Similarly, we predict greater species richness and composition during the wet season compared to the dry season, as well as differences in species composition based on the road type (roads or trails).

## 2. Results

### 2.1. Species Composition

After analyzing the plant species composition with all three datasets using DCA according to the position, zone, and season, we only observed discrepancies in species composition between the zones and the position relative to the roadside edge.

#### 2.1.1. Zones

Based on the ten zones where we worked for three years (Figure 1a), we could discriminate three groups according to their species composition: Taro, Tremesana, and the rest of the zones.

In the case of the LZ-67 road (four zones monitored during one year) (Figure 1b), species composition in the Yaiza and Taro1 zones slightly differed from the rest.

Examining the results of the fourteen zones over one year (Figure 1c), we distinguished three groups: Yaiza and Centro Visitantes differed from Montaña Rajada, Valle Tranquilidad and Altos Timanfaya, and these two groups from the other zones.

#### 2.1.2. Position

In the case of the 10 sampled zones over 3 years (Figure 2a), the plant species composition according to different zones was similar in the middle and interior transects, but differed in the border transect. For the species composition located in the LZ-67 road with four zones monitored for one year (Figure 2b), the separation between groups was not so clear. For the fourteen zones over a year (Figure 2c), the discrimination was even less evident. There did not seem to be any clear difference between the species composition at the border and the middle and the interior.

The remaining factors (season, year, and road type) were analyzed with DCA, but the variables did not show discrimination for any of the datasets (Figure A1 and Figure A2).

### 2.2. Species Richness

Based on the inventories carried out, a total of 139 species were recorded (Table A1). In 68 transects of the interior roads of the national park and the Mazo and Tremesana trails over a period of three years, 115 species were identified. We found higher number of species at the border zones of Taro and Islote, followed by the trails of Tremesana and Mazo (Table A2), especially during the winters of 2021 and 2023. After recording 34 transects of LZ-67 over the course of a year, we identified 121 species (Table A3).

Of the total 102 transects, some of them, such as *Polycarpaea divaricata*, *Forsskaolea angustifolia*, and *Rumex lunaria*, were recurrent in all zones, regardless of the seasonality. Every year, after the rains in the winter season, the number of herbaceous species recorded was high, while in the summer season, only shrubby and some herbaceous species at the edge of the roads could survive (Table A2 and Table A3).

Species richness was analyzed with linear mixed-effect models for all zones, road positions (border, middle, and interior), and seasons (summer and winter) for three years for the ten zones, for one year for the four zones of the LZ-67, and for one year for all the fourteen zones (Table 1). The best model according to the AIC for the datasets of the ten and fourteen zones was the interaction of the three factors (Table 1a,b), while for the dataset for the four zones of the LZ-67 road, it was the addition of these three factors (Table 1c).

Regarding the effect of season, in the ten zones inventoried for three years (Table 2a), there were no differences between summer and winter in the richness of species at the border. Islote displays the highest species richness at the border compared to the middle and interior, both in winter (two-fold increase) and in summer (six times as much). In Altos Timanfaya, Montaña Rajada, Valle Tranquilidad, and Manto Virgen, there were no differences in species richness between the border, middle, and interior, neither in summer nor in winter. Also, the Manto Virgen zone had no differences in species richness by position or season. In the rest of the nine zones, there was a greater effect of position in winter than in summer, showing a higher species richness at the border in winter compared to the middle and interior, while this situation only happened in summer at Islote (Tukey post hoc test, *p* < 0.05). Therefore, roads had a higher effect on species composition in winter in the Timanfaya National Park and Los Volcanes Natural Park. Of all the zones, only Calderas Quemadas, Manto Virgen, and Montaña Rajada reported no differences between summer and winter in species richness between the middle and the interior position (Tukey post hoc test, *p* ≥ 0.05).

Regarding the four zones of the LZ-67 road (Table 2b), in all of them there was a higher species richness at the border than in the middle and interior, both in summer and winter, as well as a higher species richness in winter (Tukey post hoc test, *p* < 0.05). Of the four zones, Taro 1 experienced the greatest increase in species richness between the border and the other two positions, almost four times more in summer and three times more in winter. Roads had the same effect in all zones with respect to position, and season had the same effect in all zones and positions (Table 2b) (Tukey post hoc test, *p* < 0.05).

To have a general overview, we analyzed separately the effect of zone, position, and season on species richness in all the roads during one year (summer 2022 and winter 2023) (Table 1c, Figure 3). We found a higher number of species at the border of the road (Figure 3a), followed by the middle and interior zones, as well as a higher value in winter compared to summer (Figure 3b) (Tukey post hoc test, (*p* < 0.05)).

When comparing all fourteen zones, the richness was higher in two of the transects on the main road LZ-67 (Taro 1 and Centro Visitantes), followed by the other two zones belonging to this main road (Echadero Camellos and Yaiza), and two zones inside the Timanfaya National Park that corresponded to the entrance to the park from the LZ-67 road (Taro and Islote) with the number of species decreasing as we went further into the protected areas (Figure 3c). Referring to the last winter season of 2023, at the border of roads we detected 60 species at Taro 1, 78 at Centro Visitantes, 50 at Echadero Camellos, 58 at Yaiza, 65 at Taro, and 58 at Islote (Table A2). By far, the two areas with the lowest species richness values were Manto Virgen and Calderas Quemadas, followed by Montaña Rajada, but these zones showed significant differences only with the zones with the highest species richness values (Taro1, Centro Visitantes, Echadero Camellos, Yaiza, and Taro). In this case, in the winter of 2023, we recorded 6 species on the border of the roads in Manto Virgen, 7 in Calderas Quemadas, and 14 species in Montaña Rajada (Table A2).

#### Species Richness of Endemic, Native, Introduced, and Invasive Species

Considering the types of species categorized by their endemicity (as described in Table A1 and Table A4a,b), we analyzed the number of species across different areas and time periods in datasets 1 and 2. Referring to dataset 1, the ten zones where we worked for three years (Figure 4a) from winter 2021 to winter 2023, the number of endemic species (*NS), the rest of the native species (N), invasive species (II), and the remaining introduced species (I) remained stable. There were significant differences between the endemic species and the rest of the native species in both winters (Tukey post hoc test, (*p* < 0.05)).

Regarding dataset 2, for the four zones monitored for one year (Figure 4b), significant differences were found between the number of endemic species (*NS) and nonendemic native species (N) and between the summer of 2022 and the winter of 2023 (Tukey post hoc test, (*p* < 0.05)), unlike between invasive introduced species (II) and the rest of the introduced (I) species through time, where no differences were found. In these four zones, the number of species increases significantly in winter compared to summer, especially among native species.

Introduced invasive species (II) did not exhibit differences in any of the datasets (Tukey post hoc test, *p* ≥ 0.05).

## 3. Discussion

Roads play a crucial role in shaping the composition and especially the number of plant species in Timanfaya National Park and Los Volcanes Natural Park, which are environmentally sensitive areas. These parks serve as prime examples of recent volcanic activity in the Canary Islands and are particularly vulnerable to human impact.

### 3.1. Species Composition

These protected natural areas exhibit high environmental microheterogeneity. Based on the ten zones where we conducted our research for three years, and according to their environmental conditions, we can differentiate Altos de Timanfaya, Mazo, or Taro from Montaña Rajada or Tremesana. The first three zones are situated in the northern part of these protected areas, with a significant moisture input that promotes vegetation colonization, while Montaña Rajada, and especially Tremesana, are in the southernmost zone, with warmer and drier conditions. The discrepancies on species composition between Taro, Tremesana, and the rest of the zones may be due not only to their environmental conditions, but also to the level of human impact they experience, as it has been shown in the case of the laurel forest in Tenerife [32]. *Helianthemum canariense*, *Astragalus solandri*, *Anacyclus radiatus*, *Emex spinosa*, and *Kickxia sagittata* are some of the species that only appear in these two zones. Taro, Tremesana, and, to a lesser extent, Islote and Mazo are zones with higher levels of vehicle, pedestrian, and bicycle traffic, which can act as vectors facilitating seed dispersal, as described in Australia, the USA, and New Zealand [3]. The rest of the zones are located along the tourist route of the national park, and are only accessible to buses with no other passage allowed. In the case of the LZ-67 road, Yaiza shares the same environmental conditions as Tremesana, zones located in the southern part of the park where plant colonization and survival are more challenging due to the higher temperatures and lower rainfall. On the other hand, in the other three zones, especially in Taro1 and Centro Visitantes, the species cover is very high. These are the zones located further north, with a greater contribution of humidity, and where there is a higher density of parked vehicles. That is, both are access zones to Timanfaya National Park in the first case, and to the Centro Visitantes in the second case. Comparing the species composition in the fourteen zones over one year, Yaiza and Centro Visitantes, which are more exposed to tourism and human impact, differ from Altos Timanfaya, Valle Tranquilidad, and Montaña Rajada, which are located within the bus tour route inside the national park, and from the rest of the zones.

The position factor had a slight influence on the species composition [37,38]. The difference in species composition between the border and the middle and interior of the roads may be largely influenced by the season, due to the emergence of annual herbs in winter, and also depending on the zone, as we mentioned previously. The position factor is visible in the ten zones sampled over a period of three years. In the case of the LZ-67 main road, differences on the species composition according to the position are not clear, although the pattern follows a similar tendency. This could be due to the fact that the species which are able to establish, even in less favorable areas (far from the border), are well adapted to survive there. This is the case of, for example, *Launaea arborescens*, *Polycarpaea divaricata*, *Rumex lunaria*, *Pelargonium capitatum*, and *Forsskaolea angustifolia*, among others. The absence of a clear edge effect in certain road transects was also described in other national parks, such as the ones in temperate forests of south–central Chile [14], where the communities are open and subjected to intense grazing and firewood collection pressure, which hinders the edge effect. When we considered the fourteen zones over the last year, this trend did not change. No major discrepancies in species composition were observed at the border, middle, and interior of the roads. It is possible that the influence of the LZ-67 zones is stronger and may mask the general trend for species composition; that is, to show differences between the border, middle, and interior of the roads.

### 3.2. Species Richness

Referring to species richness, and based on our data analysis, in the datasets related to the ten and fourteen zones, season affects species richness in a given zone and position, just as the zone also affects the number of species in each position. Depending on the road locations and their environmental conditions, they will have a different effect on species richness, as well as varying effects according to the season. Thus, all zones are favored after the winter rains, leading to a clear increase in species richness. When we only account for the effect of the position, we found a higher number of species at the roadside compared to the middle and interior of all roads. Species such as *Chamaesyce serpens*, *Melilotus sulcatus*, *Cyperus capitatus*, *Emex spinosa*, or *Plantago afra* appear only at the border of all the sampled roads, with the first two species being introduced to the island. Some species seem to be favored by the altered conditions as a consequence of the road and might be transported by the wheels of the vehicles and the air in which they move [20]. Also, species richness is higher in winter than in summer, and in particular, some winters of the study were particularly wet for the island of Lanzarote. In 2021, the rainfall was intense, and in 2023, several pulses of water fell for several months, favoring the formation of annual grass lawns in the first meters from the road edge.

Although species richness is not only influenced by the impact of roads, but also by the environmental conditions of the different zones, this first factor seems to be particularly relevant, as has been shown in other studies around the world [15,16,20]. In Centro Visitantes, Taro 1, Taro, and Echadero Camellos, there is a high daily traffic of vehicles, with tourists waiting to take a bus and visit Timanfaya National Park, these being the areas with the highest number of species, respectively (with only Centro Visitantes and Taro 1 showing significant differences from the rest of the zones). Almost daily, the volume of people is so high that they have to wait their turn parked in these areas. In this way, both vehicles and people act as seed-dispersal vectors that reach these road edges [15]. The same happens on the trails of Tremesana and Mazo. Although access to the first is a bit more restrictive, people, pets, bicycles, and vehicles transit these zones daily, generating an impact on seed dispersal and favoring plant colonization. Tremesana is more arid, so perhaps plant colonization is more challenging even with the effect of the road. In Mazo, the impact is much greater, and the edge effect in winter is remarkable, with a species richness three times higher on the border than in the middle and interior. In contrast, in Tremesana, the difference in species richness between the border, middle, and interior is less than 50%. There was no difference in species richness between trails and roads due to the small number of trails compared to roads, making it difficult to discern their effect from other factors. Yaiza is also highlighted for its high number of species, because, despite being the driest zone, as mentioned before, the first sampling of this zone was taken in a roundabout close to a village and showed a high number of species, with some urban-associated species such as *Opuntia dillenii*, *Chenopodium murale*, or *Nicotiana glauca*. Thus, the distance to urban centers also plays a role [39]. However, Manto Virgen and Calderas Quemadas were the zones with the lowest species richness. Manto Virgen is dominated by pahoehoe or corded lava flows, which make it difficult for species to settle and survive. Only in the cavities and roadsides with cinder do some species such as *Umbilicus gaditanus* or *Polycarpaea divaricata* emerge, respectively. In this area, we found some shrubs, such as *Launaea arborescens* or *Rumex lunaria*. The greater impact is evident in the difference in species richness between LZ-67 and the other zones. This difference clearly shows the influence that a highly trafficked main road can generate compared to roads that, for the most part, have controlled traffic due to their belonging to a protected natural area. We have identified on this road halophytic species, such as *Cyperus capitatus*, and other species from ecosystems that are not related to this volcanic landscape. According to oral sources, this is likely due to the transit of trucks carrying sand from other parts of the island. Additionally, the highest presence of invasive species was found in these 34 transects from the highly trafficked road. Neotropical plant species, including *Opuntia* sp. or *Agave* sp., are also present. These invasive species were introduced in the Canary Islands for various purposes, including marking land boundaries and properties, and their expansion has been facilitated by these anthropogenic corridors or roads in the archipelago [40,41]. Nevertheless, out of the 16 existing introduced species recorded in the present study, only 6 of them were invasive (*Agave americana*, *Atriplex semibaccata, Nicotiana glauca, Opuntia dillenii*, *Rumex lunaria*, and *Pelargonium capitatum*) (Table A1). Although *Rumex lunaria* is listed in some catalogues as endemic to the Canary Islands [42], it is considered a translocated native species on Lanzarote [43] and with an invasive behavior on the island and in the National Park. Furthermore, with the exception of *Rumex lunaria* and *Pelargonium capitatum*, the number and cover of invasive species has remained practically unchanged over these three years. This fact explains the high level of conservation that these protected natural areas have. Even so, over these three years, we have recorded 30 new species (Bernardos, M. et al., in prep) that had not been previously cataloged in the guides published for the Timanfaya National Park [44,45], and one of these is invasive (*Atriplex semibaccata*). This remarks on the ongoing need for careful monitoring and the establishment of appropriate management practices to address the potential impacts of invasive species and to maintain the ecological balance of the park, just as occurs in other national parks [14].

Our study highlights the important role that roads play in accelerating ecological succession processes, by modifying the environmental conditions that facilitate the spread and establishment of species in these sensitive volcanic areas, as well as the need to adopt management practices that take these processes into account in order to preserve these natural protected areas. The rapid expansion of the road network has aimed to improve accessibility and mobility across the islands, connecting urban centers, tourist destinations, and natural areas. However, it is essential to carefully consider the environmental impacts associated with this extensive road development, particularly in ecologically sensitive areas such as Lanzarote. The model of use of Timanfaya National Park seems to be effective. Restricting and controlling access to a significant portion of the area for tourists and keeping major roads away from protected zones are good conservation measures. Nevertheless, traffic in certain access areas such as Taro should be regulated to prevent tourist congestion while waiting for their visit. It is crucial to plan effective tourism management to conserve our protected natural spaces to the best extent possible. Balancing the need for transportation infrastructure with sustainable and environmentally conscious practices is crucial to mitigate the potential negative effects on ecosystems and promote the long-term conservation of natural resources. Just as the Netherlands, Australia, and the United States are leaders in road ecology [15], this study could serve as a valuable input for decision-making in the management of protected and touristic areas, and to raise awareness of the impact of road systems on other volcanic islands in the world, which are more susceptible to biological invasions.

## 4. Materials and Methods

### 4.1. Study Site

Our study zones were located on roads in Timanfaya National Park and Los Volcanes Natural Park on Lanzarote Island, Canary Islands, Spain. These protected areas represent the recent volcanism in the Canary Islands, with eruptions in the 18th and 19th centuries [46] that resulted in the present landscape with more than 25 craters in an area of a few squared kilometers. This is a space dominated by recent volcanic formations (lava flows, tephra cones, and other pyroclastic products) where soil formation is scarce or nonexistent, limiting the colonization of vascular plant species [47]. It is a lunar-like landscape, primarily colonized by lichens, some annual herbs, sparse perennials, and a few shrub species [46].

The island of Lanzarote has a warm desert climate, characterized by low rainfall, mild winters, and dry summers [48]. The average annual rainfall was 141 mm from 1970 to 2020, with some periods of 3–4 years experiencing 60 mm. The average annual temperature in the Timanfaya National Park was 19.35 °C from 1991 to 2020, similar to the island mean annual temperature (19.56 °C) [49]. The atmospheric humidity hovers around 60%, and the sunshine duration is, on average, eight hours per day. During the summer, the frequency of NNE winds is 40–45%, whereas in winter, it is between 13 and 30% [48].

### 4.2. Design of the Experiment

We analyzed the species composition by determining number of species (species richness) and species cover of road and trail transects in fourteen different zones of Timanfaya National Park and Los Volcanes Natural Park (Figure 5). The monitoring was carried out at two different times of the year: the dry season (September) and the wet season (March), over three years (2020–2023).

We analyzed a total of 68 transects for three years (dataset 1) in Timanfaya National Park (including Ruta de Los Volcanes, Chinero, Montaña Rajada, and Taro) and Los Volcanes Natural Park (including the Mazo and Tremesana trails). In addition, during the last two campaigns (September 2022 and March 2023), 34 new transects from the road leading to the National Park (LZ-67) were monitored (dataset 2), departing from the Yaiza roundabout, going through the national park’s camel dropoff and the Taro area to the Visitor Center. Therefore, for two campaigns, a total of 102 transects were studied in 14 different environmental zones (dataset 3) (Table 3, Figure 5).

We worked on two types of roads: asphalted roads (roads) and dirt tracks (trails). Out of the fourteen zones analyzed, only two of them were trails (Mazo and Tremesana) located in the Los Volcanes Natural Park. The first one is a public trail that is used by walkers, vehicles, and bicycles. The second one is a private trail, where access is restricted to vehicles entering the national park and to the owners of fruit trees within the park. Walkers can also access and walk the trail to the boundary where the national park begins. The twelve remaining zones comprise only roads. Along the road LZ-67 (Figure 5), which is a major via, with very fluid vehicle traffic, that provides access to the national park and connects the populations of Yaiza, Manchas Blancas, and Tinajo, we analyzed 34 transects. The rest of the zones (67 transects) are located within the Timanfaya National Park, and all, except Taro and Islote, are only accessible by buses than run every day for a maximum of eight-and-a-half hours, providing tourists with a tour of the volcanic landscape. Other types of traffic are banned in the interest of the conservation of the park.

We collected some features for each of the 102 transects: zone, number of transects, elevation (m a.s.l.), mean slope (°Sex.), exposure, mean annual temperature (°C), mean annual precipitation (mm), road type, and number of years sampled (Table 3). In addition, other abiotic factors not listed in the table were recorded: substrate type and coverage, date of roadside management (if there were signs of recent cutting of vegetation), and trash presence.

Each transect consisted of a 50 m length and 5 m width area, divided in six 1 m wide plots (three plots on each side of the road), with 250 m of uninventoried road between transects, for a total of approximately 30 km tracked (Figure 6).

The first plot or “border” covered the first meter adjacent to the road (0–1 m); the second plot or “middle” covered the meter between 2 and 3 m away from the border; the third plot or “interior” covered the meter between the 4 and 5 m from the edge of the road. Therefore, a 1 m gap between each plot was excluded from the inventory. In this ecosystem, at a distance of 4–5 m from the edge, we considered that the effect of the road is minimal, and the natural composition of species would exist. Beyond this distance, the impact of road-related factors, such as pollutants, altered abiotic conditions, and air movements that can disperse seeds, diminishes, allowing for a more undisturbed environment [34].

In each of the six plots per section, we noted all the native and non-native species of vascular flora present (richness), in addition to estimating their cover, following Van Der Maarel’s scale: 1, traces; 2, <1%; 3, 1–2%; 4, 2–5%; 5, 5–10%; 6, 10–25%; 7, 25–50%; 8, 50–75%; 9, >75%; 10, 100% [50].

Furthermore, we analyzed this species richness according to the types of endemicity described in Table A1, resulting in the following:

*NS: endemic species;

N: native species. We grouped the three types of native species (NS, NP, and NO) into a single category, N;

II: introduced invasive species;

I: introduced species. We grouped the rest of introduced species in this category.

In this analysis, we worked with datasets 1 (ten zones monitored over three years) and 2 (four zones monitored for one year) to investigate whether the number of species in each category varied over time depending on whether the zones were more protected (interior zones of dataset 1) or more exposed to anthropogenic impact (zones along LZ-67 in dataset 2).

Regarding dataset 1, we only considered data from the first and last winters, specifically the winters of 2021 and 2023. This decision was based on the fact that most annual herb species were recorded in winter rather than summer. In the case of dataset 2, we could only use data from that year, specifically from the summer of 2022 and the winter of 2023.

### 4.3. Statistical Analyses

All the analysis were done using the R computational language (R v.4.2.0) [51]. We performed all the analyses separately with three datasets covering: (1) 10 zones monitored for three years, (2) 4 zones belonging to the LZ-67 road monitored for one year, and (3) the 14 zones monitored for one year. To analyze the influence of position (border–middle–interior), zone and season on species composition, we carried out a DCA (detrended correspondence analysis) using vegan package [52]. DCA is a useful technique for analyzing high-dimensional datasets and interpreting species distribution across different sites or environmental gradients. To determine the effects of the position, zone, and season on the richness of species (number of species), we developed linear mixed-effects models applying the lmer function from the lme4 package [53]. We established position, zone, and season as fixed factors (both with interaction and addition term) and transect as random factor. Model *p*-values were obtained using bootstrapping with the function PBmodcomp from the pbkrtest package [54]. To assess the influence of endemicity and time on species richness (number of species), we also constructed linear mixed-effects models using the lmer function from the lme4 package [53]. We specified endemicity and time as fixed factors (with interaction term), while considering transect, position, and zone as random factors. Model selection was based on the lowest AIC. Richness of species was log transformed to meet the assumptions of homoscedasticity and normality of model residuals. Differences between groups were detected with a Tukey HSD post hoc test. The *p*-value was settled to 0.05.

## 5. Conclusions

Our study emphasizes the impact of roads in promoting the distribution of both native and non-native species in volcanic areas. The borders of roads provide a suitable environment for plant species to grow and spread due to the buffer or transitional area created by the road edge, which provides a favorable microclimate for the establishment of plant species in an otherwise arid environment. The alteration of soil conditions as a consequence of the existence of the road itself and the effect of the vehicles bringing seeds and generating air movements also favor this colonization.

Environmental microheterogeneity allows for differences in species composition between the border and the interior of the roads. There are differences due to the effect of roads in the more heavily trafficked zones compared to the more controlled ones: no edge effect in species composition is observed along the busy road LZ-67 (more homogeneous zones, often extending the border effect towards the interior); a more pronounced edge effect is evident in the interior roads of the national and natural park (zones more sensitive to the impact of roads).

The road does have an effect on the number of species, this being higher at the border. We detected a higher species richness in the zones along the LZ-67 road. Not only the position, but also the zone and season have effects on the number of species. There is a seasonal effect on species richness across positions, with higher richness observed in winter and at the border of roads. Understanding these patterns assists in park management, such as controlling invasive species at road edges, for instance.

The analysis of the road-edge effect is useful to understand how infrastructure causes environmental variability, changes the colonization capacity, and favors the entry of species. During the years of the research, we detected a moderate expansion of invasive species. Thus far, Timanfaya National Park has exhibited effective management practices that need to be sustained over time. It is crucial to maintain responsible tourism management and control access to protected areas to ensure the proper conservation of the space. Roads are features that need to be carefully planned, constructed, and maintained while considering their relevant ecological implications.

## Figures and Tables

**Figure 1 plants-12-03568-f001:**
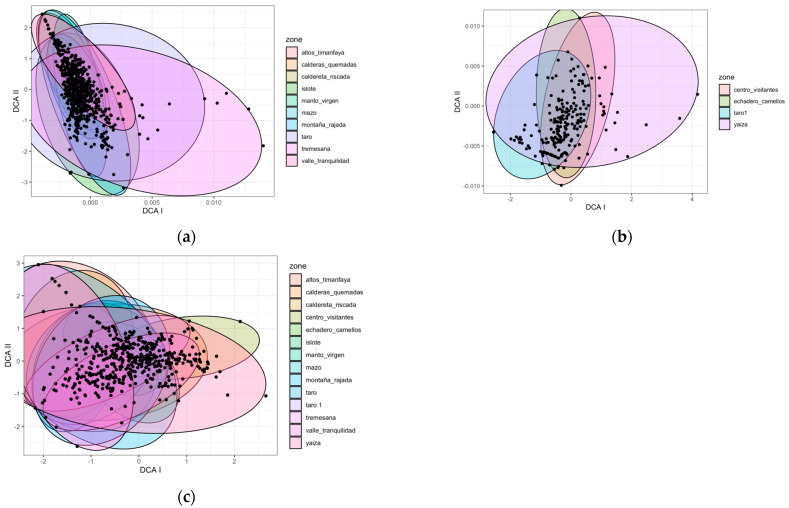
DCA characterizing species composition according to zones under different environmental conditions in (**a**) the ten zones sampled for three years, and (**b**) the case of the LZ-67 road with four zones monitored for one year, and (**c**) all fourteen zones for one year. Each point on the graph represents a position within a transect for each season and year.

**Figure 2 plants-12-03568-f002:**
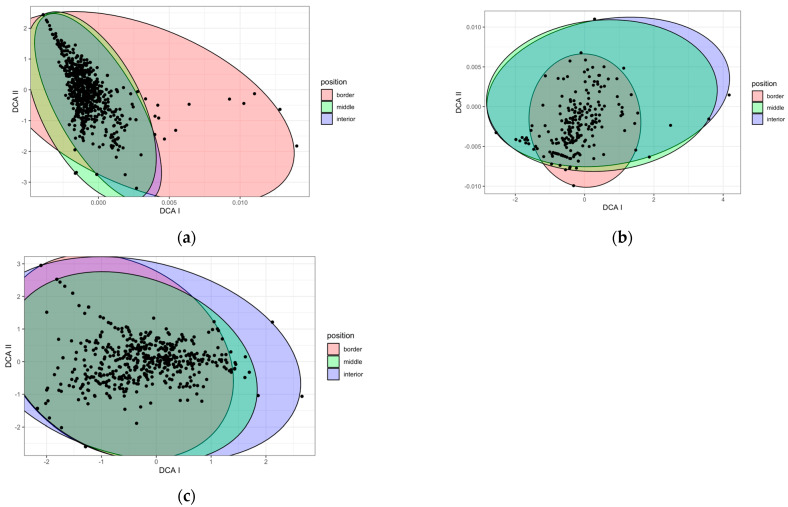
DCA based on species composition according to the position in respect to the road border in (**a**) the ten zones sampled for three years, (**b**) the case of LZ-67 road with four zones monitored for one year, and (**c**) all fourteen zones for one year. Each point on the graph represents a position within a transect for each season and year.

**Figure 3 plants-12-03568-f003:**
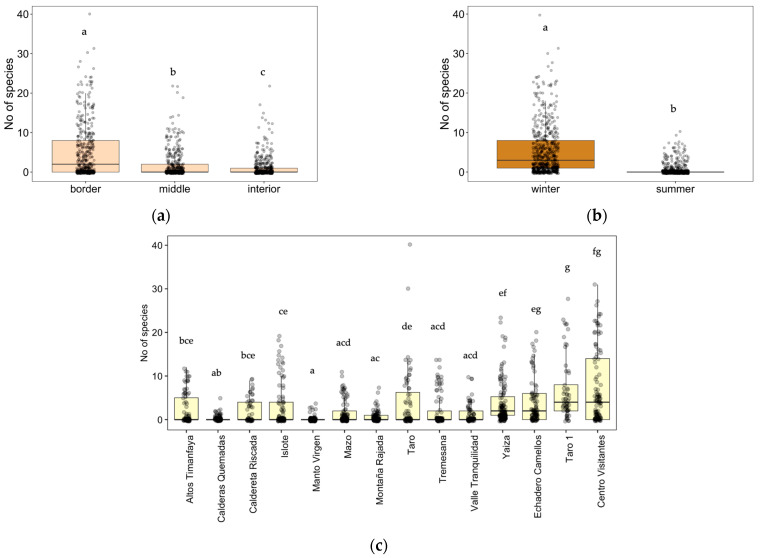
(**a**) Effect of the position (distance to the road: border, middle, or interior) on the number of species, (**b**) effect of the season (winter, summer) on the number of the species, (**c**) and effect of zones (Altos Timanfaya, Calderas Quemadas, Caldereta Riscada, Islote, Manto Virgen, Mazo, Montaña Rajada, Taro, Tremesana, Valle Tranquilidad, Yaiza, Echadero Camellos, Taro 1, and Centro Visitantes) following linear mixed-effect models with position, zone, season, and transect as random factors and Tukey post hoc test, *p* < 0.05. Different letters indicate significant differences between groups.

**Figure 4 plants-12-03568-f004:**
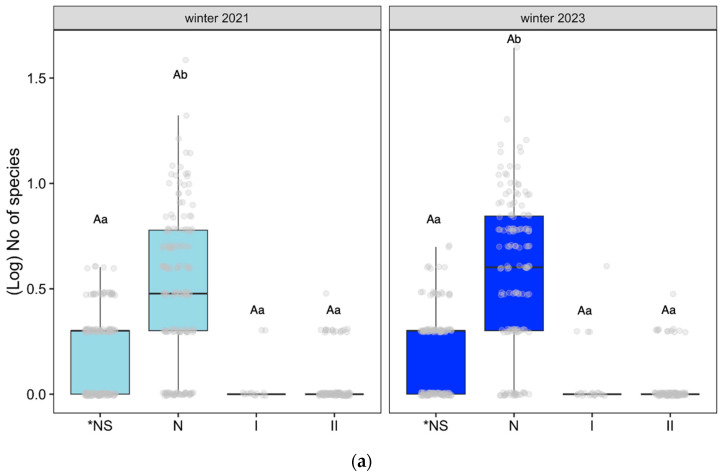
(**a**) Effect of the endemicity (*NS: endemic species; N: the rest of the native species; II: invasive introduced species; I: the rest of the introduced species;) on the number of species on dataset 1: Altos Timanfaya, Calderas Quemadas, Caldereta Riscada, Islote, Manto Virgen, Mazo, Montaña Rajada, Taro, Tremesana, and Valle Tranquilidad (only using data from the first and last winters); (**b**) Effect of the endemicity (*NS: endemic species; N: the rest of the native species; II: invasive introduced species; I: the rest of the introduced species) on the number of species on dataset 2: Yaiza, Echadero Camellos, Taro 1, and Centro Visitantes, following linear mixed-effect models with position, zone, and transect as random factors and Tukey post hoc test, *p* < 0.05. We only analyzed differences in endemicity within each specific time and changes over time within the same dataset. Different small letters indicate significant differences between time, and capital letters show significant differences in endemicity.

**Figure 5 plants-12-03568-f005:**
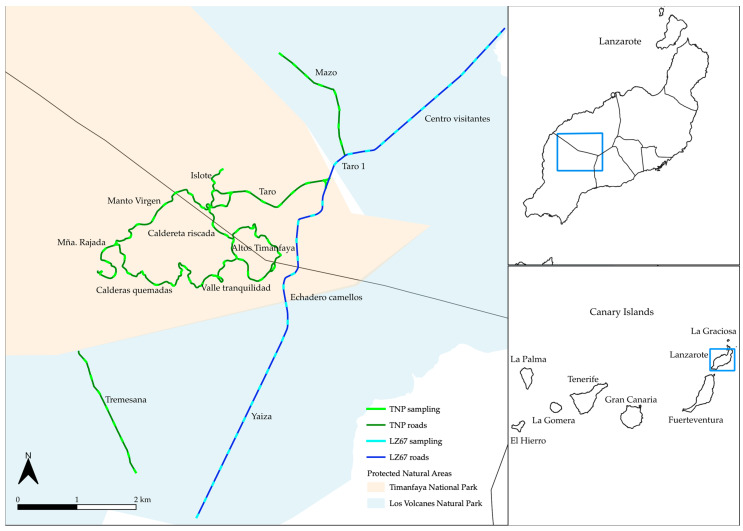
Location of the roads and trails sampled. For three years, we monitored the roads and trails colored in dark green. The light green transects correspond to each of the transects. The dark blue road with the light blue transects were analyzed for one year.

**Figure 6 plants-12-03568-f006:**
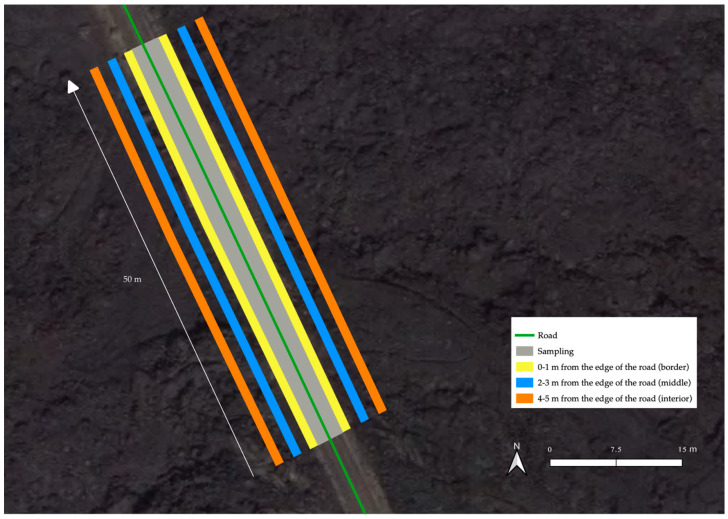
Transect sampling design. The six transects of one section are shown according to the distance from the roadside (0–1 m; 2–3 m; 4–5 m) on both the right and left sides of the road (three transects at each side).

**Table 1 plants-12-03568-t001:** Summary of linear mixed models, showing the effects of zones (dataset 1: Altos Timanfaya, Calderas Quemadas, Caldereta Riscada, Islote, Manto Virgen, Mazo, Montaña Rajada, Taro, Tremesana, and Valle Tranquilidad; dataset 2: Yaiza, Echadero Camellos, Taro 1, and Centro Visitantes; dataset 3: all zones), position (border, middle, and interior), and season (summer, winter) on species richness. We applied five models to the three separate datasets: (**a**) the 10 zones inventoried for 3 years; (**b**) the 4 zones inventoried for 1 year; (**c**) the 14 areas inventoried for 1 year. P and AIC values are shown (in bold, significant *p* values and the best model according to the lowest AIC). Significant *p* values: ‘***’ 0.001; ‘*’ 0.05.

	Model	AIC	*p*
**(a)**	Log(richness) = position + zone + season	3634.177	**0.000999 *****
	**Log(richness) = position × zone × season**	**3538.346**	**0.000999 *****
	Log(richness) = zone	4533.766	**0.000999 *****
	Log(richness) = position	4554.275	**0.000999 *****
	Log(richness) = season	3942.127	**0.000999 *****
**(b)**	**Log(richness) = position + zone + season**	**533.1070**	**0.000999 *****
	Log(richness) = position × zone × season	541.8778	**0.000999 *****
	Log(richness) = zone	1065.6377	**0.016983 ***
	Log(richness) = position	905.8038	**0.000999 *****
	Log(richness) = season	848.4138	**0.000999 *****
**(c)**	Log(richness) = position + zone + season	2149.202	**0.000999 *****
	**Log(richness) = position × zone × season**	**1778.268**	**0.000999 *****
	Log(richness) = zone	3108.800	**0.000999 *****
	Log(richness) = position	3024.786	**0.000999 *****
	Log(richness) = season	2515.595	**0.000999 *****

**Table 2 plants-12-03568-t002:** Species richness of the 14 sampling zones according to inventory position (border, middle, and interior) and season (winter and summer). Results for the (**a**) ten zones monitored for three years and (**b**) the four areas of the LZ-67 sampled for one year. Given are means ± SD. Different small letters indicate significant differences between position, and capital letters show significant differences in season, following linear mixed-effect models and Tukey post hoc test, *p* < 0.05.

			Species Richness	
			Season	
	Zones	Position	Summer	Winter
**(a)**	Altos Timanfaya	Border	2.1 ± 1.0 aA	6.1 ± 3.8 aB
		Middle	1.3 ± 0.5 aA	5.4 ± 4.0 aB
		Interior	1.5 ± 0.8 aA	5.3 ± 4.4 aB
	Calderas Quemadas	Border	1.8 ± 0.8 aA	2.8 ± 1.5 aB
		Middle	1.0 ± 0.0 aA	2.3 ± 1.3 bA
		Interior	1.0 ± 0.0 aA	1.7 ± 0.8 bA
	Caldereta Riscada	Border	2.9 ± 1.0 aA	6.9 ± 2.5 aB
		Middle	2.0 ± 1.3 aA	5.3 ± 2.2 abB
		Interior	1.3 ± 0.5 aA	4.4 ± 2.3 bA
	Islote	Border	6.2 ± 2.1 aA	28.5 ± 6.1 aB
		Middle	0.9 ± 1.5 bA	14.1 ± 10.2 bB
		Interior	1.0 ± 2.2 bA	11.3 ± 8.3 bB
	Manto Virgen	Border	5.3 ± 2.3 aA	18.6 ± 5.6 aA
		Middle	0.5 ± 0.5 aA	5.0 ± 3.2 aA
		Interior	0.8 ± 0.9 aA	3.2 ± 1.5 aA
	Mazo	Border	5.0 ± 3.2 aA	15.5 ± 4.7 aB
		Middle	2.0 ± 1.0 aA	5.5 ± 3.7 bB
		Interior	1.4 ± 0.5 aA	3.1 ± 2.0 bA
	Montaña Rajada	Border	1.6 ± 0.9 aA	2.8 ± 1.4 aB
		Middle	1.3 ± 0.6 aA	1.9 ± 1.6 aA
		Interior	1.0 ± 0.0 aA	2.3 ± 2.0 aA
	Taro	Border	2.0 ± 1.4 aA	7.8 ± 4.9 aB
		Middle	1.8 ± 1.2 aA	3.4 ± 3.0 bB
		Interior	1.8 ± 1.5 aA	2.6 ± 2.7 bB
	Tremesana	Border	1.4 ± 0.6 aA	3.6 ± 2.7 aB
		Middle	1.2 ± 0.4 aA	2.2 ± 1.4 bB
		Interior	1.1 ± 0.4 aA	2.9 ± 2.1 bA
	Valle Tranquilidad	Border	5.5 ± 4.9 aA	16.1 ± 13.5 aB
		Middle	2.0 ± 1.3 aA	6.5 ± 4.7 aB
		Interior	1.8 ± 0.9 aA	4.5 ± 3.4 aB
**(b)**	Yaiza	Border	3.5 ± 2.4 aA	14.8 ± 7.1 aB
		Middle	0.9 ± 1.1 bA	8.8 ± 9.3 bB
		Interior	1.0 ± 1.3 bA	3.8 ± 6.2 bB
	Echadero Camellos	Border	5.3 ± 2.3 aA	18.6 ± 5.6 aB
		Middle	0.5 ± 0.5 bA	5.0 ± 3.2 bB
		Interior	0.8 ± 0.9 bA	3.2 ± 1.5 bB
	Taro 1	Border	7.8 ± 1.3 aA	29.8 ± 3.6 aB
		Middle	2.0 ± 1.6 bA	9.8 ± 4.3 bB
		Interior	2.4 ± 2.1 bA	8.8 ± 8.2 bB
	Centro Visitantes	Border	6.2 ± 2.1 aA	28.5 ± 6.1 aB
		Middle	0.9 ± 1.5 bA	14.1 ± 10.2 bB
		Interior	1.0 ± 2.2 bA	11.3 ± 8.3 bB

**Table 3 plants-12-03568-t003:** Environment characterization of the sampling zones. Given are means ± SD.

Zone	No. Transects	Elevation (m a.s.l.)	Mean Slope (°Sex.)	Exposure	Mean Annual Temperature (°C) [48]	Mean Annual Precipitation (mm) [48]	Road Type	No. Years Sampled
Altos Timanfaya	7	430.4 ± 13.7	11.9 ± 5.8	N	18.3 ± 0.1	174.0 ± 2.8	Road	3
Calderas Quemadas	6	293.2 ± 20.6	11.0 ± 4.9	SE	18.9 ± 0.1	157.1 ± 2.4	Road	3
Caldereta Riscada	4	356.3 ± 25.6	11.4 ± 5.3	SO	18.7 ± 0.1	159.8 ± 2.3	Road	3
Islote	8	317.2 ± 11.5	10.4 ± 4.4	N	18.9 ± 0.0	156.9 ± 1.7	Road	3
Manto Virgen	6	288.2 ± 18.0	9.2 ± 4.5	N	19.0 ± 0.1	155.5 ± 1.3	Road	3
Mazo	8	263.8 ± 25.9	2.7 ± 1.6	SO	19.0 ± 0.1	158.8 ± 4.2	Trail	3
Montaña Rajada	8	278.6 ± 33.0	13.2 ± 9.5	N	19.0 ± 0.1	157.4 ± 4.0	Road	3
Taro	6	300.9 ± 4.3	3.2 ± 2.0	N	18.9 ± 0.0	158.5 ± 1.7	Road	3
Tremesana	8	196.0 ± 10.3	4.3 ± 2.4	E	19.4 ± 0.1	148.8 ± 0.6	Trail	3
Valle Tranquilidad	6	341.6 ± 26.4	9.3 ± 3.7	N	18.7 ± 0.1	163.0 ± 3.2	Road	3
Yaiza	11	225.7 ± 25.0	4.0 ± 1.6	E	19.3 ± 0.1	154.0 ± 2.2	Road	1
Echadero Camellos	8	333.2 ± 20.7	5.4 ± 2.9	E	18.8 ± 0.1	166.6 ± 1.6	Road	1
Taro 1	5	307.8 ± 17.5	6.5 ± 7.3	N	18.8 ± 0.1	161.8 ± 1.8	Road	1
Centro Visitantes	10	294.9 ± 12.4	3.1 ± 2.6	N	18.7 ± 0.0	167.3 ± 2.4	Road	1

## Data Availability

New data were created or analyzed in this study. We can provide data under request.

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
