# Peer review of "Road Impact on Plant Colonization in the Arid Timanfaya National Park"

_plants, 2023, doi:10.3390/plants12203568_

Round 1

Reviewer 1 Report

The authors investigated the effect of road on plant colonization in the arid Timanfaya National Park and give the management practices for natural protected areas. The results highlight the complex relationship between the effect of roads and zones with different environmental conditions on species composition and richness, which are meaningful for clarify the plant response to human activities and provide important theoretical basis for ecology management. The topic of the study fits within the scope of the journal, and the study provides some interesting new data points.

The authors should highlight the innovation and add the hypothesis of their paper. Writing/clarity requires extra attention. The followings are the detailed comments:

1. Line 16, 109, 130, please add the word “plant” before “species composition”.

2. Line 91, To understand the influence of zones on....?

3. P in line 171 and table 1, use in italic or non-italic.

4. Line 173, the effect of season on the position?

5. Table 2, revise “,” between the number as “.”?

6. Line 390, We sampled the species composition by determining number of species, revise “sampled” as “analyzed” or “investigated”?

7. Line 427, revised “Abiotic characterization” as “Environment characterization”?

8. Please add the subtitle in the discussion section.

9. Line 567, Rev. Ecol. Syst., and line 596, Urban Plan. should be used in italic.

Author Response

The authors investigated the effect of road on plant colonization in the arid Timanfaya National Park and give the management practices for natural protected areas. The results highlight the complex relationship between the effect of roads and zones with different environmental conditions on species composition and richness, which are meaningful for clarify the plant response to human activities and provide important theoretical basis for ecology management. The topic of the study fits within the scope of the journal, and the study provides some interesting new data points.

The authors should highlight the innovation and add the hypothesis of their paper. Writing/clarity requires extra attention. The followings are the detailed comments:

  1. Line 16, 109, 130, please add the word “plant” before “species composition”. Done
  2. Line 91, To understand the influence of zones on....? Done
  3. P in line 171 and table 1, use in italic or non-italic. Done
  4. Line 173, the effect of season on the position? Done
  5. Table 2, revise “,” between the number as “.”? Done
  6. Line 390, We sampled the species composition by determining number of species, revise “sampled” as “analyzed” or “investigated”? Done
  7. Line 427, revised “Abiotic characterization” as “Environment characterization”? Done
  8. Please add the subtitle in the discussion section. Done
  9. Line 567, Rev. Ecol. Syst., and line 596, Urban Plan. should be used in italic. Done

Thank you very much for the comments and corrections; we have incorporated and addressed all the changes you have suggested.

Reviewer 2 Report

Authors deal with interesting topic important for nature conservation. The study presents new data but some interpretations are not based on data presented and I miss some important results.

 Abstract is informative, but it contains also note about the presence of native and non-native species, which is not a part of the results presented in the article.

Introduction is well written, no specific comments from my side.

Results are mostly oriented to the species richness in relation to the position/distance to the road. I miss some analysis focusing on the presence of native and non-native species with a special emphasis to endemic and invasive plant species.

Discussion is adequate, comparing data of the study with relevant literature.

Materials and Methods are well described, methods used are adequate.

Conclusions are supported by the results, with the exception of the last paragraph, where expansion of non-native species is mentioned – I miss any data about this effect.

Apendix A  - tab A1 – in legend, mark for endemic genus is presented but it is not used in the table.

Author Response

Authors deal with interesting topic important for nature conservation. The study presents new data but some interpretations are not based on data presented and I miss some important results.

 Abstract is informative, but it contains also note about the presence of native and non-native species, which is not a part of the results presented in the article. Done

Introduction is well written, no specific comments from my side.

Results are mostly oriented to the species richness in relation to the position/distance to the road. I miss some analysis focusing on the presence of native and non-native species with a special emphasis to endemic and invasive plant species.  Done

Discussion is adequate, comparing data of the study with relevant literature.

Materials and Methods are well described, methods used are adequate.

Conclusions are supported by the results, with the exception of the last paragraph, where expansion of non-native species is mentioned – I miss any data about this effect.  Done

Apendix A  - tab A1 – in legend, mark for endemic genus is presented but it is not used in the table. Done

Thank you so much for your contributions. We have taken them into account to include the corresponding analyses and changes in the article.

Reviewer 3 Report

This research emphasizes the need to recognize the effects of road construction on the types of species that inhabit a given area. It shows that building roads can increase the likelihood of invasive species entering the region, while also enhancing the ability of native species to colonize new areas. The paper contains detailed and significant findings, and I recommend it for publication with the following changes: 1) Rearrange the chapters in the standard format for publishing: introduction, materials and methods, results, discussion, and conclusion. 2) Abbreviate any excessive text, images, and tables in the appendix and move them to supplementary material.

 Minor editing of the English language is required

Author Response

This research emphasizes the need to recognize the effects of road construction on the types of species that inhabit a given area. It shows that building roads can increase the likelihood of invasive species entering the region, while also enhancing the ability of native species to colonize new areas. The paper contains detailed and significant findings, and I recommend it for publication with the following changes: 1) Rearrange the chapters in the standard format for publishing: introduction, materials and methods, results, discussion, and conclusion. 2) Abbreviate any excessive text, images, and tables in the appendix and move them to supplementary material.

Thank you very much for your assessments and for the comments. Regarding 1), we have followed the order proposed by the Plants MDPI journal template. According to point 2), the content of the appendix is the supplementary material.